# Antibody Dynamics Simulation—A Mathematical Exploration of Clonal Deletion and Somatic Hypermutation

**DOI:** 10.3390/biomedicines11072048

**Published:** 2023-07-20

**Authors:** Zhaobin Xu, Qingzhi Peng, Weidong Liu, Jacques Demongeot, Dongqing Wei

**Affiliations:** 1Department of Life Science, Dezhou University, Dezhou 253023, China; pqz0317@163.com; 2Department of Physical Education, Dezhou University, Dezhou 253023, China; liuweidong1971@126.com; 3Laboratory AGEIS EA 7407, Team Tools for e-Gnosis Medical, Faculty of Medicine, University Grenoble Alpes (UGA), 38700 La Tronche, France; 4School of Life Sciences and Biotechnology, Shanghai Jiao Tong University, Shanghai 200030, China; dqwei@sjtu.edu.cn

**Keywords:** positive selection, negative selection, somatic hypermutation, self-tolerance, antibody, aging, mathematical modeling

## Abstract

We have employed mathematical modeling techniques to construct a comprehensive framework for elucidating the intricate response mechanisms of the immune system, facilitating a deeper understanding of B-cell clonal deletion and somatic hypermutation. Our improved model introduces innovative mechanisms that shed light on positive and negative selection processes during T-cell and B-cell development. Notably, clonal deletion is attributed to the attenuated immune stimulation exerted by self-antigens with high binding affinities, rendering them less effective in eliciting subsequent B-cell maturation and differentiation. Secondly, our refined model places particular emphasis on the crucial role played by somatic hypermutation in modulating the immune system’s functionality. Through extensive investigation, we have determined that somatic hypermutation not only expedites the production of highly specific antibodies pivotal in combating microbial infections but also serves as a regulatory mechanism to dampen autoimmunity and enhance self-tolerance within the organism. Lastly, our model advances the understanding of the implications of antibody in vivo evolution in the overall process of organismal aging. With the progression of time, the age-associated amplification of autoimmune activity becomes apparent. While somatic hypermutation effectively delays this process, mitigating the levels of autoimmune response, it falls short of reversing this trajectory entirely. In conclusion, our advanced mathematical model offers a comprehensive and scholarly approach to comprehend the intricacies of the immune system. By encompassing novel mechanisms for selection, emphasizing the functional role of somatic hypermutation, and illuminating the consequences of in vivo antibody evolution, our model expands the current understanding of immune responses and their implications in aging.

## 1. Introduction

The immune system serves as a pivotal defense mechanism against exogenous pathogenic invaders. Of particular significance is the intricate functioning of adaptive immunity within higher organisms, wherein pathogen-specific responses are orchestrated while simultaneously preserving self-tolerance to prevent autoimmune manifestations under physiological conditions [1,2,3]. Recent years have witnessed extensive experimental investigations into the maturation and differentiation processes governing T-cells and B-cells, shedding light on the intricacies underlying their development [4,5,6,7,8,9,10].

These studies have elucidated the fundamental role played by self-antigens in orchestrating the developmental trajectory of nascent immune cells. Self-antigens act as guiding signals, allowing thymocytes and bone marrow-derived B-cells to undergo positive selection, ensuring the generation of a diverse repertoire capable of recognizing antigens while maintaining self-tolerance [11,12,13]. This intricate process establishes the foundation for immune competence while safeguarding against aberrant immune reactions directed towards self-structures. Additionally, these investigations have unveiled the phenomenon of negative selection or clonal deletion during lymphocyte maturation. This tightly regulated process eliminates autoreactive clones within the developing lymphocyte population, thwarting the emergence of self-reactive effector cells that could provoke deleterious autoimmunity. By selectively eliminating autoreactive lymphocytes, the immune system ensures the preservation of immunological self-tolerance, preventing the onset of harmful autoimmune diseases. Taken together, the exploration of T-cell and B-cell ontogeny offers invaluable insights into the delicate balance between effective immune responses and self-tolerance maintenance. It underscores the sophistication of adaptive immunity, highlighting the intricate interplay of molecular checkpoints and selection processes involved in shaping the immune repertoire. These discoveries offer a deeper understanding of the immune system’s ability to mount potent defenses against pathogens while evading self-directed attacks, furthering our comprehension of immune homeostasis and fostering potential interventions for immunological disorders.

However, there is currently a dearth of a systematic model that comprehensively investigates the intricacies involved in studying the specificity of antibody-mediated immune responses. While several mathematical models have been proposed to examine virus–host interactions in immunology, these models often fail to encompass the underlying physical mechanisms fundamental to immunological processes [14,15,16,17,18,19,20]. The majority of these models rely heavily on fitting clinical data and suffer from limited applicability due to their narrow scope of mathematical formulas. Moreover, existing models tend to primarily emphasize the stimulatory impact of pathogenic microorganisms on antibody production, with scant attention given to the essential processes of positive and negative selection that occur within the immune system itself. To genuinely comprehend the true mechanism at play in phenomena influenced by multiple factors rather than a singular principle, it becomes imperative to adopt a systematic approach through comprehensive mathematical modeling. Consider the developmental process wherein immune cells undergo both positive and negative selection. Self-antigen binding facilitates immune cell maturation, while excessive binding frequently leads to clone deletion, resulting in negative selection. Relying solely on rudimentary experimental observations, we can merely ascertain the simultaneous occurrence of positive and negative selection during immune cell development. However, elucidating the precise causes behind this phenomenon and deciphering the transitions between these processes remains elusive without the aid of an encompassing mathematical model.

To more systematically investigate the manifestation of adaptive immunity, our proposed model integrates the concept of somatic hypermutation of germinal center B-cells. This inclusion allows for an in-depth exploration of its effects on immune cell proliferation and self-antigen tolerance. By venturing beyond the conventional boundaries of immunological research, our model strives to unravel the complexities of immune responses and provide quantitative insights into various dynamic issues associated with the immune system. The establishment of a robust mathematical model holds paramount importance in unraveling the specificity of antibody-mediated immune responses. By capturing the intricate physical mechanisms and considering multiple influencing factors concurrently, such models foster a nuanced understanding of immunological processes. This not only enables the quantitative analysis and prediction of immune responses to diverse pathogens, but also lays the foundation for future advancements in research and medical interventions within the domain of immunology.

In addition, we present a novel theory known as the immuno-driven theory of human aging, which distinguishes itself from conventional theories concerning immunological aging [21,22]. Our theory posits that long-term homeostasis within the human immune system is unattainable, primarily due to the dynamic nature of antibodies and their influence on the isotypes of mature B-cells. As individuals age, antibodies tend to exhibit heightened autoimmune characteristics. Although somatic hypermutation can mitigate this process to some extent, it cannot fundamentally reverse the prevailing trend. The progressive increase in autoimmune activity serves as a catalyst for the aging process, encompassing not only the overall deterioration of bodily functions but also the aging of the immune system itself. This phenomenon heralds a paradigm shift, highlighting the crucial role of immunological dynamics in shaping the aging trajectory of an organism. By elucidating the intricate relationship between immunosenescence and autoimmunity, our theory broadens the scope of understanding regarding human aging. It underscores the significance of incorporating immunological factors into comprehensive models that explore the complexities of aging. Through this interdisciplinary approach, we aim to unravel the underlying mechanisms governing age-related immune dysfunction and shed light on potential interventions aimed at mitigating the deleterious effects of immunological aging. By recognizing the impact of antibody dynamics and heightened autoimmune activity, our theory offers fresh insights into the molecular and cellular processes driving the aging phenotype. Furthermore, it serves as a springboard for future research endeavors, propelling advancements in the field of immunology and facilitating the development of novel therapeutic strategies to counteract age-related immune dysregulation.

A brief organization is delineated as follows: Section 2 presents the formulation of two distinct models. The first model, a simplistic framework devoid of consideration for the effects of somatic hypermutation, elucidates the underlying mechanisms behind B-cell clonal deletion. Additionally, it provides an explanation for the counterintuitive phenomenon wherein antibodies exhibiting robust binding affinity to exogenous antigens undergo rapid proliferation instead of deletion. Subsequently, the second model, encompassing somatic hypermutation, elucidates the proactive role of somatic hypermutation in immunological processes. Its influence extends beyond expediting the resolution of autoimmune disorders, as it facilitates the generation of a myriad of high-affinity antibodies imperative for combating pathogenic incursions. The apparent paradoxical effects are effectively reconciled and explicated through the utilization of the second model. Finally, employing the second model, an exploration into the temporal evolution of the organism’s antibody repertoire was undertaken. It was discerned that antibodies tend to evolve towards an augmented binding affinity to self-antigens. Although somatic hypermutation demonstrates a pronounced palliative effect, it does not fundamentally reverse this trajectory. Henceforth, this postulation is denoted as the immune-driven senescence hypothesis.

## 2. Materials and Methods

### 2.1. A Simplified Mathematical Model of Antibody Dynamics (Model 1)

In our model, we introduce variables to represent the different components involved in the immune response to viral antigens. The number of antibody–antigen complexes is denoted by the variable *x*, the total number of antibodies is denoted by variable *y*, and the number of viruses is denoted by variable *z*. Our model encompasses six distinct processes outlined as follows:

The first reaction describes the proliferation or replication of the virus, which occurs with a reaction constant denoted as $k_{1}$.

The second reaction represents the binding interaction between the virus and antibody. It involves a forward reaction constant ($k_{2}$) and a reverse reaction constant ($k_{-2}$), reflecting the affinity between the virus and the antibody.

The third reaction involves the removal of the antibody–virus complex. This process is characterized by a reaction constant denoted as $k_{3}$.

The fourth reaction signifies the induction of new antibodies through the antibody–virus complex. It occurs with a kinetic constant represented as $k_{4}$. In immunology, these virus–antibody complexes are initially located on the surface of B-cells since antibodies are primarily produced by B-cells and become attached to their plasma membranes. Subsequently, B-cells proceed to digest and decompose the viral antigens into shorter peptides. The antigen-presenting cells, in this case represented by B-cells, present the viral peptide (the antigen) to T-cells. The physical interaction entails the binding of B-cells to T-cells. The T-cells then process and respond to the presented antigens; if the antigens are foreign, they secrete strong signaling molecules to promote the proliferation or division of the B-cells attached to them. Consequently, both B-cells and the antibodies generated by these B-cells undergo proliferation. Conversely, if the presented peptides are self-derived (homogenous to self), the proliferation signals emitted by helper T-cells are relatively weak, providing limited proliferative potential to both themselves and the attached B-cells [23].

The fifth reaction reflects the degradation of the virus, which occurs at a constant rate denoted as $k_{5}$.

The sixth reaction represents the degradation of antibodies, characterized by a rate constant represented as $k_{6}$. Those six reactions are illustrated in Figure 1.

By incorporating these six processes into our model, we aim to gain a comprehensive understanding of the dynamics involved in the immune response to viral antigens. This sophisticated representation enables us to explore the intricate interplay between viruses, antibodies, and various immune cell populations, contributing to advancements in immunological research and potentially guiding the development of novel therapeutic strategies against viral infections.

We established the following equations to represent the proliferation process of antibodies, which was derived from our previous publication [24].
(1)$$\frac{dx}{dt}=k_{2}yz-k_{-2}x-k_{3}x,$$
(2)$$\frac{dy}{dt}=k_{-2}x-k_{2}yz+k_{4}x-k_{6}y,$$
(3)$$\frac{dz}{dt}=k_{-2}x-k_{2}yz-k_{5}z+k_{1}z.$$

Those processes are used to model the interaction between the antibody and pathogen. For self-antigens, *z* cannot proliferate by itself but would remain as a constant. It would be consumed by the binding of its corresponding antibodies but would be quickly replenished by the regeneration process. Therefore, the interaction of antibodies and self-antigens can be represented as the following equations:(4)$$\frac{dx}{dt}=k_{2}yz-k_{-2}x-k_{3}x,$$
(5)$$\frac{dy}{dt}=k_{-2}x-k_{2}yz+k_{4}x-k_{6}y,$$
(6)$$\frac{dz}{dt}=0.$$

Here *x* denotes the antibody–self-antigen complex, *y* denotes antibodies, and *z* denotes self-antigens. The proliferation signals generated by its corresponding T helper cell are much weaker compared to foreign antigens, with a much smaller $k_{4}$ value in this case. Model 1 is utilized to simulate the mechanism of B-cell clonal deletion, and the specific simulation results and parameter selections are presented in Section 3.1.

### 2.2. A Kinetic Model Considering Somatic Hypermutations in Antibodies (Model 2)

To comprehensively study the impact of somatic hypermutations on the evolution of antibodies within the body, it is crucial to establish a reliable mutation model that accurately simulates the effects of mutations on antibody binding affinity. Initially, somatic hypermutations generate random mutations that primarily affect the binding affinity in the immune process [25]. Some mutations can enhance the binding affinity with the corresponding antigen, while others may decrease it. Additionally, certain mutations have a minimal impact on binding affinity.

Biophysical statistical experience suggests that when starting with a template with poor binding affinity, any point mutation is likely to enhance the binding affinity. Conversely, when using a starting template sequence with excellent binding affinity, the majority of point mutations will lead to a significant decrease in binding affinity. In our model, the binding affinity is represented by the parameters $k_{2}$ and $k_{-2}$. To roughly estimate the impact of antibody mutations on their binding affinity, we utilized the latest research data from Miller et al. [26]. Around 356 antigen–antibody complexes were selected from the SAbDab database, and their binding affinity information was extracted. The distribution of binding affinities roughly follows a normal distribution with a mean ***K***_d_ value of −9 and a variance of 1, which is shown in Figure 2.

By constructing such a mutation model and simulating it with real data, we can gain a better understanding of the mechanisms through which somatic hypermutations influence antibody evolution. This knowledge provides valuable insights for antibody engineering and immunological research.

When considering the phenomenon of somatic hypermutations in antibodies, it becomes apparent that antibodies with different binding affinities undergo interconversion. In order to capture this interconversion explicitly, a more refined approach is needed. The continuous distribution of antibody–antigen binding affinities in Model 2 feedback poses challenges for simulation using ordinary differential equations (ODEs). To address this, a discretization strategy is employed, whereby the system is discretized into a finite number of affinity units. In this particular case, the discretization is performed with a granularity of nine distinct antibody affinity units.

Each antibody unit is capable of transforming into its adjacent unit according to a conversion probability that follows a normal distribution. The degree of discretization plays a crucial role in determining the accuracy of the results obtained. Higher levels of discretization yield more precise outcomes but concurrently increase the complexity of interpreting the results and computational requirements. This discretization approach allows for a comprehensive exploration of the dynamics of antibody affinity evolution. By incorporating probabilistic conversions between discrete affinity states, the model captures the inherent variability and plasticity of the immune response. Such a specialized and academic approach offers insights into the underlying kinetics of somatic hypermutations and can aid in understanding the factors influencing antibody diversity and antigen recognition in the immune system.

As shown in Figure 2, during somatic hypermutation in somatic cells, antibody *i* can transition to a higher-affinity antibody *i* + 1, a lower-affinity antibody *i* − 1, or undergo a mutation that does not affect its affinity, denoted as antibody *i*. While these mutations can alter the sequence of the antibody, they have a minimal impact on its binding affinity. The probability of somatic hypermutation is represented as p. The rate at which antibody *i* transitions to itself is given by (1 − p)$k_{4}x_{i}$ + p$S_{i,i}k_{4}x_{i}$, where $S_{i,i}$ represents the area under the curve for region *i*.

The rate at which antibody *i* transitions to *i* + 1 is given by p$S_{i-1,i}k_{4}x_{i}$, where $S_{i-1,i}$ denotes the probability of conversion from antibody *i* to *i* − 1, equivalent to the area under the curve for the right side of region *i*. Similarly, the rate at which antibody *i* transitions to *i* − 1 is governed by p$S_{i+1,i}k_{4}x_{i}$, where $S_{i+1,i}$ represents the area under the curve for the left side of region *i*. It is important to note that $S_{i,i}$ can be obtained through the integration of the normal distribution.

For instance, in our study, we divided the antibodies into nine distinct binding affinity classes. In the event of somatic hypermutation, a specific scenario occurred where the sixth antibody (highlighted in green) had a probability of 0.6914 to transform into the fifth antibody, which has a stronger binding capacity (highlighted in red). Furthermore, there was a probability of 0.2417 for the transformed antibody to persist as an antibody with similar binding affinity. Additionally, there was a probability of 0.0668 for the transformed antibody to further convert into the seventh antibody (highlighted in blue). It is important to note that the summation of these probabilities is equal to 1.

Thus, the dynamics of antibody affinity changes during somatic hypermutation can be quantitatively described using these rate equations. This specialized and academic translation provides a deeper understanding of the underlying mechanisms involved in antibody affinity evolution and contributes to the broader field of immunology research.

When the antigenic substance is the self-originated, this process can be represented by the following equation:(7)$$\frac{dx_{i}}{dt}=k_{2i}y_{i}z-k_{-2i}x_{i}-k_{3}x_{i},$$
(8)$$\frac{dy_{i}}{dt}=k_{-2i}x_{i}-k_{2i}y_{i}z+(1-p)k_{4}x_{i}+pS_{i}k_{4}x_{i}+pS_{i,i+1}k_{4}x_{i+1}\;+pS_{i,i-1}k_{4}x_{i-1}-k_{6}y_{i},$$
(9)$$\frac{dz}{dt}=0.$$

Among these factors, $k_{4}$ is exclusively determined by the antigen’s properties and remains unaffected by the antibody binding affinity. $k_{6}$ denotes the inherent decay rate of antibodies, which remains consistent across different antibody types. Additionally, $k_{3}$ signifies the rate at which antibody complexes are cleared and is invariant among various antibody species. Notably, different antibodies exhibit distinct kinetic association coefficients $k_{2i}$ and dissociation coefficients $k_{-2i}$. In our study on the simulation of somatic hypermutation in B-cells, we utilized a set of 9 antibodies characterized by specific association rate constants, denoted as $k_{2\_1}$ to $k_{2\_9}$ ($k_{2\_1}\;$ = 10−^13^, $k_{2\_2}\;$ = 10^−12^, $k_{2\_3}\;$ = 10^−11^, $k_{2\_4}\;$ = 10^−10^, $k_{2\_5}\;$ = 10^−9^, $k_{2\_6}\;$ = 10^−8^, $k_{2\_7}\;$ = 10^−7^, $k_{2\_8}\;$ = 10^−6^, $k_{2\_9}\;$ = 10^−5^). It was postulated that the dissociation constants $k_{-2}$ remained constant for all antibodies at a uniform value of 10^−18^. As a result, the equilibrium constants (***K***_d_ values) corresponding to each antibody were determined to be ($k_{d\_1}\;$ = 10^−5^, $k_{d\_2}\;$ = 10^−6^, $k_{d\_3}\;$ = 10^−7^, $k_{d\_4}\;$ = 10^−8^, $k_{d\_5}\;$ = 10^−9^, $k_{d\_6}\;$ = 10^−10^, $k_{d\_7}\;$ = 10^−11^, $k_{d\_8}\;$ = 10^−12^, $k_{d\_9}\;$ = 10^−13^), which has a good match to statistical reports [25].

## 3. Results

### 3.1. Possible Physical Mechanisms of Clonal Deletion

Clonal deletion of B-cells is a critical regulatory process in the immune system, aimed at eliminating potentially pathogenic B-cells to maintain immune balance and self-tolerance. This process is referred to as B-cell negative selection.

In the bone marrow, hematopoietic stem cells undergo a series of differentiation stages to generate mature B-cells. During this process, B-cells undergo scrutiny based on their reactivity to self-antigens and foreign antigens, selectively retaining cells with a normal immune function that do not pose a risk of self-damage. Specifically, clonal deletion involves two main mechanisms: self-antigen recognition and immune regulatory signaling.

Self-antigen recognition refers to the binding of antigens by the B-cell receptor (BCR) present on the surface of B-cells, leading to an immune response. In early development, B-cells in the bone marrow encounter a variety of self-antigens and employ mechanisms of self-antigen recognition to determine their interaction. If the affinity between a B-cell’s BCR and self-antigens is excessively high, resulting in hyperactive immune reactions, it may lead to autoimmune diseases. To prevent such scenarios, these overly reactive B-cells undergo negative selection. During negative selection, multiple regulatory mechanisms are initiated to eliminate B-cells with potential harm. One mechanism involves exposing these B-cells to high concentrations of self-antigens, inducing them towards apoptotic (cell death) pathways. Additionally, immune regulatory cells such as regulatory T-cells also participate by secreting regulatory cytokines to trigger B-cell apoptosis or restrict their survival and proliferation capabilities, thereby suppressing the expansion of abnormal clones.

Our model proposes a possible new mechanism for clonal deletion. The fate of B-cells primarily depends on two aspects: the binding affinity between their encoded antibodies and the corresponding antigens, and the stimulatory capacity of the resulting peptides from antigen cleavage on the corresponding helper T-cells.

For exogenous antigens, the peptide sequences formed after antigen cleavage often possess strong stimulating effects on Th cells. This stimulation prompts T-cells to secrete a large amount of growth factors, which in turn promotes the division and proliferation of the corresponding Th cells. As B-cells are spatially linked to these Th cells, the growth factors also strongly stimulate the proliferation of B-cells, leading to the amplification of the corresponding antibodies. Although antigen-bound B-cells can be easily recognized and eliminated by other immune cells such as NK cells or the complement system, the proliferative effect of Th cells surpasses the rate of elimination. Hence, the overall dynamics still exhibit a rapid expansion trend, as depicted in Figure 3a.

Specifically, when the peptide’s stimulatory effect on Th cells is weak (*k*_4_ = 0.1) for self-antigenic substances, high-affinity antibodies (*k*_2_ = 10^−8^) are cleared more rapidly than low-affinity antibodies (*k*_2_ = 10^−9^). This is because high-affinity antibodies more readily form antigen–antibody complexes with self-antigens, leading to their further clearance by the immune system, accompanied by the elimination of the corresponding B-cells. For self-antigenic substances, their peptide sequences generated after antigen cleavage have a very weak stimulatory effect on Th cells, thus are unable to induce Th cells to secrete a large amount of growth factors for proliferation. However, due to the high binding affinity between the antibody and the antigen, the corresponding B-cells quickly cover the antigenic substances, accelerating their apoptosis. At this stage, there is also binding between these B-cells and Th cells, causing the physical co-elimination of both cell types. The overall dynamics demonstrate a rapid deletion trend of antibodies.

Distinct from pathogenic microorganisms, self-antigenic substances lack the ability to undergo rapid expansion and are not easily eliminated in a short period of time. Their concentration can remain relatively stable over an extended period. Therefore, in our simulation, we consider their concentration to be constant. Conversely, as shown in Figure 3b, for exogenous microbial infections, the higher the binding affinity of the antibodies, the faster they proliferate. This ensures the rapid selection and proliferation of high-affinity antibodies by our immune system.

Additionally, we observe that antibodies with high self-binding activity undergo rapid deletion. Many experimental studies have reported that clonal deletion is not a complete elimination of antibodies with strong self-binding activity, especially during the early stages of B-cell development [27,28,29]. Clonal deletion is thus considered to be a progressive process. Therefore, we lean towards the theory that the dominant factor driving clonal deletion are the antibody dynamics reflected in this model, rather than additional special mechanisms to handle B-cells with high self-binding activity.

The choice of parameters has a profound impact on the simulation outcomes. Typically, parameter values are determined through literature mining or experimental fitting. Regrettably, specific experimental data pertaining to the clonal deletion of B-cells in the context of our model remains elusive, thus diminishing the model’s veracity. Nevertheless, an advantage of our model is its qualitative explication of certain phenomena, wherein consistent trends emerge within a given parameter range.

Regarding Figure 3a, the initial concentration of antigen–antibody complexes (denoted as *x*(0)) is assumed to be zero. The initial concentration of antibodies (denoted as *y*(0)) is approximately 10^5^, while the initial concentration of endogenous antigenic material (denoted as *z*(0)) is approximately 3 × 10^7^. These initial parameter values draw inspiration from our prior publications [24,30]. Importantly, the overall trend exhibits low sensitivity to the precise selection of initial concentrations. Specifically, when the initial antibody concentration (*y*(0)) resides in the order of 10^5^ and the initial endogenous antigenic material concentration (*z*(0)) resides in the order of 10^7^, a consistent trend emerges whereby antibodies exhibiting strong affinity for self-antigens are promptly eliminated. For instance, when contemplating somatic hypermutation in somatic cells, Figure 4 and Figure 5 stipulate a selection of *z*(0) at 2 × 10^6^. Similar principles guide the selection of other parameters. Employing available data on the kinetics of antigen–antibody binding, the dissociation coefficient $k_{-2}$ assumes an extremely minute value, namely 10^−18^. Correspondingly, the forward binding coefficient $k_{2}$ is constrained to the range elucidated in Section 2.2, spanning from 10^−5^ to 10^−13^, based on binding affinity considerations and the equilibrium constant ***K***_d_. A larger value signifies a heightened binding affinity. In our scenario, a value of 10^−8^ is elected to denote robustly binding antibodies, while 10^−9^ signifies weakly binding antibodies in Figure 3a. The parameter $k_{3}$ captures the rate at which antigen–antibody complexes are cleared. Its value ought to reside between 0 and 1, with larger values indicating swifter clearance rates. Notably, this rate must significantly exceed that of natural antibody decay. Consequently, we adopt a value of 0.02 for the natural decay rate of antibodies $k_{6}$, while the clearance rate of antigen–antibody complexes assumes a value of 0.5. The parameter $k_{4}$ depicts the stimulating effect of antigen–antibody complexes on antibody regeneration. Herein, a diminutive value of 0.1 is chosen as the coefficient governing self-antigen–antibody complex-induced antibody regeneration. For exogenous antigens, this stimulation coefficient should be substantially augmented.

**Figure 3 biomedicines-11-02048-f003:**
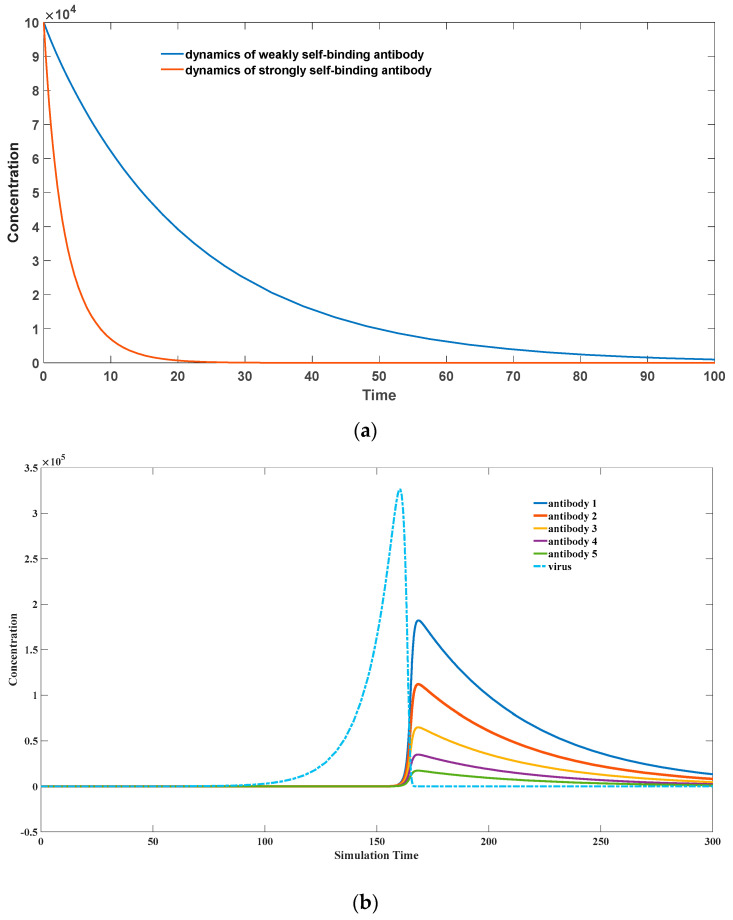
(**a**) A possible scenario of clonal deletion. The parameter sets we used are: *z*(0) = 3 × 10^7^, x(0) = 0, y(0) = 10^5^, $k_{2}$ = 10^−8^
for a strongly binding antibody, $k_{2}$ = 10^−9^ for a weakly binding antibody, $k_{-2}$ = 10^−18^, $k_{3}$ = 0.5, $k_{4}$ = 0.1, $k_{6}$ = 0.02. (**b**) Selection of good binding antibodies toward pathogen infection (Adopted from our previous publication [24]). The parameter sets we used are: *x*(0) = 0, *y*(0) = 1, *z*(0) = 1, *w* = 1, $k_{1}=0.1,k_{2}$ = 10^−5^, $k_{-2}$ = 10^−14^, $k_{3}$ = 1, $k_{4}$ = 2, $k_{5}$ = 0.02, $k_{6}$ = 0.02, for antibody 1; $k_{2}$ = 9 × 10^−6^, $k_{-2}$ = 9 × 10^−15^ for antibody 2; $k_{2}$ = 8 × 10^−6^, $k_{-2}$ = 8 × 10^−15^ for antibody 3; $k_{2}$ = 7 × 10^−6^, $k_{-2}$ = 7 × 10^−15^ for antibody 4; $k_{2}$ = 6 × 10^−6^, $k_{-2}$ = 6 × 10^−15^ for antibody 5.

### 3.2. Somatic Hypermutation Accelerates Self-Immune Response Elimination

Somatic hypermutation (SHM) is a pivotal process occurring in B-cells during their maturation within the germinal centers of secondary lymphoid organs, including the spleen and lymph nodes. It plays a fundamental role in generating high-affinity antibodies and promoting immune responses.

The primary outcome of somatic hypermutation is the induction of point mutations within the variable region of immunoglobulin genes encoding the antigen-binding site of antibodies. This diversifies the repertoire of antibodies generated by B-cells, enabling recognition and binding to a wide range of antigens with enhanced specificity.

Somatic hypermutation involves the activation-induced cytidine deaminase (AID) enzyme, which targets the variable region of immunoglobulin genes. AID converts cytosine residues in the DNA sequence to uracil through deamination [31,32]. The presence of uracil triggers DNA repair mechanisms such as base excision repair (BER) and mismatch repair (MMR), which introduce random nucleotide substitutions during the repair process. These substitutions contribute to the generation of a diverse set of antibody variants.

The introduction of somatic mutations creates a pool of B-cells with varying affinities for the antigen. Through a process known as affinity maturation, B-cells with higher-affinity antibodies are preferentially selected for survival and proliferation, while those with lower-affinity or non-functional antibodies undergo apoptosis. During affinity maturation, B-cells presenting antibodies with greater affinity for the antigen are more likely to receive survival signals from T follicular helper (Tfh) cells and follicular dendritic cells [23]. These cells provide essential survival and proliferation signals, including cytokines and antigen presentation, respectively, ensuring the expansion and persistence of B-cell clones producing high-affinity antibodies. By iteratively undergoing rounds of somatic hypermutation, selection, and clonal expansion, the immune system progressively enhances its capacity to produce antibodies with increasing affinity for the antigen. This iterative process allows B-cells to finely tune their immune responses and generate highly specific and effective antibodies against invading pathogens.

It is important to note that somatic hypermutation is tightly regulated, as excessive or uncontrolled mutations can lead to the production of autoantibodies, contributing to the development of autoimmune diseases. Consequently, multiple mechanisms exist to prevent the generation of self-reactive antibodies, including clonal deletion, receptor editing, and regulatory T-cell control. Somatic hypermutation represents a critical mechanism in humoral immune responses. It introduces random nucleotide substitutions within immunoglobulin genes, resulting in the generation of a diverse antibody repertoire. Through selection and clonal expansion, B-cells producing high-affinity antibodies are favored, leading to a more efficient immune response [33,34,35]. However, the precise regulation of somatic hypermutation is necessary to prevent deleterious autoimmune reactions.

In recent years, it has been discovered that somatic hypermutation, a type of event involving extensive genetic recombination in immune cells, occurs continuously and differs from germ-line genetic recombination. Even mature B-cells, when proliferating and generating offspring, display antibody sequences that are different from the parental generation [36,37,38]. This ongoing somatic hypermutation not only increases antibody diversity to combat pathogen infections but also plays a role in rapidly eliminating self-immune reactions. Experimental evidence has shown that mice lacking somatic hypermutation exhibit strong long-term autoimmune responses when injected with autoreactive antibodies produced by B-cells. In contrast, mice with normal somatic hypermutation capabilities can swiftly suppress these intense self-immune reactions [39]. This indicates the crucial role of somatic hypermutation in maintaining self-tolerance.

Our model confirms the positive influence of somatic hypermutation in preserving self-tolerance. To construct this model, we initially developed a model to assess the impact of mutations on binding affinity. Accurate calculation of the effects of mutations on binding affinity can be obtained by statistically analyzing the changes in binding affinity caused by numerous point mutations. Such statistics can be derived from the experimental data or estimated by computational methods based on the calculated binding energy after mutation. For simplicity, in our study, we utilized an existing database and roughly estimated that the binding coefficients between antigens and antibodies follow a normal distribution with a mean of −9 and a variance of 1. This distribution assumes that most mutations tend to move towards moderate binding affinities. When the binding affinity is stronger than the average value, the probability of mutations leading to even stronger binding gradually decreases. Conversely, when the binding affinity is weaker than the average value, the probability of mutations resulting in stronger binding increases gradually.

We investigated the reciprocal conversion and kinetic characteristics of antibodies with different binding affinities using a discrete approach. In our model, as described in Section 2.2, we examined the dynamics of antibody populations based on their binding affinities. From Figure 4, it can be observed that when there is a strong self-binding antibody present, somatic hypermutation accelerates the elimination of these self-binding antibodies. In this scenario, mutations tend to drive the antibodies towards lower binding affinities.

Specifically, as the probability of somatic hypermutation decreases, antibodies with higher binding activities (antibodies 6, 7, 8) undergo substantial proliferation, leading to an overall increase in the antibody levels. Consequently, this may result in the occurrence of severe autoimmune diseases. In the presence of significant somatic hypermutation, as shown in Figure 4a, antibodies with various binding activities eventually diminish. This represents the gradual disappearance of inflammatory reactions associated with autoimmune responses, which is consistent with experimental evidence. Additionally, the total quantity of antibodies declines more rapidly compared to the absence of somatic hypermutation.

In contrast, for exogenous pathogenic microbial antigens, somatic hypermutation accelerates the generation of a large quantity of high-affinity antibodies to combat infections. This scenario is depicted in Figure 5. Specifically, in Figure 5a, for the low binding activity antibody 1 (*k*_2_ = 10–13), without somatic hypermutation, it is unable to undergo amplification and generate more high-affinity antibodies. Moreover, it fails to achieve rapid proliferation and gradually becomes eliminated. In Figure 5b, in the presence of somatic hypermutation, the low activity antibody 1 can evolve into various high-affinity antibodies and undergo rapid expansion. Therefore, somatic hypermutation plays a crucial and positive role in the immune system’s response to exogenous microbial infections.

The main cause of this distinction lies in the difference in the peptide sequences presented by antigen-presenting cells (APCs). Exogenous peptide sequences often elicit strong T-helper cell responses, whereas endogenous self-peptide sequences typically induce weaker T-helper cell responses. In our model, this distinction is mainly manifested by the different values of *k*_4_.

**Figure 4 biomedicines-11-02048-f004:**
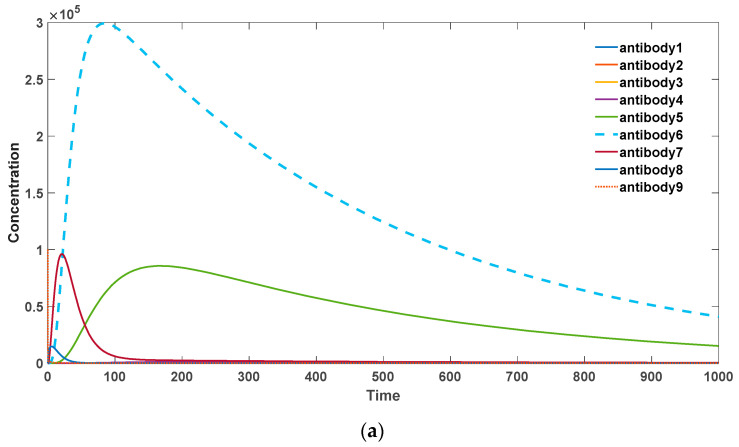
(**a**) Antibody dynamics with SHM rate equal to 0.4. (**b**) Antibody dynamics with SHM rate equal to 0.3. (**c**)
Antibody dynamics with SHM rate equal to 0.2. (**d**) Antibody dynamics with SHM rate equal to 0.1. The parameter sets we used are: *z*(0) = 2 × 10^6^, $x_{1}\left(0\right)$ = 0, $x_{2}\left(0\right)$ = 0, $x_{3}\left(0\right)$ = 0, $x_{4}\left(0\right)$ = 0, $x_{5}\left(0\right)$ = 0, $x_{6}\left(0\right)$ = 0, $x_{7}\left(0\right)$ = 0, $x_{8}\left(0\right)$ = 0, $x_{9}\left(0\right)$ = 0, $y_{1}\left(0\right)$ = 0, $y_{2}\left(0\right)$ = 0, $y_{3}\left(0\right)$ = 0, $y_{4}\left(0\right)$ = 0, $y_{5}\left(0\right)$ = 0, $y_{6}\left(0\right)$ = 0, $y_{7}\left(0\right)$
= 0, $y_{8}\left(0\right)$ = 0, $y_{9}\left(0\right)$ = 10^5^, 
$k_{2\_1}$ = 10^−13^, $k_{2\_2}$ = 10^−12^, 
$k_{2\_3}$ = 10^−11^, 
$k_{2\_4}$ = 10^−10^, $k_{2\_5}$ = 10^−9^, 
$k_{2\_6}$ = 10^−8^, $k_{2\_7}$ = 10^−7^, 
$k_{2\_8}$ = 10^−6^, 
$k_{2\_9}$ = 10^−5^,
$k_{-2}$ = 10^−18^, 
$k_{3}$ = 0.5, $k_{4}$ = 0.6, $k_{6}$ = 0.02, *p* 
= 0.4 in (**a**), *p* = 0.3 in (**b**), *p* = 0.2 in (**c**), *p* = 0.1 in (**d**).

**Figure 5 biomedicines-11-02048-f005:**
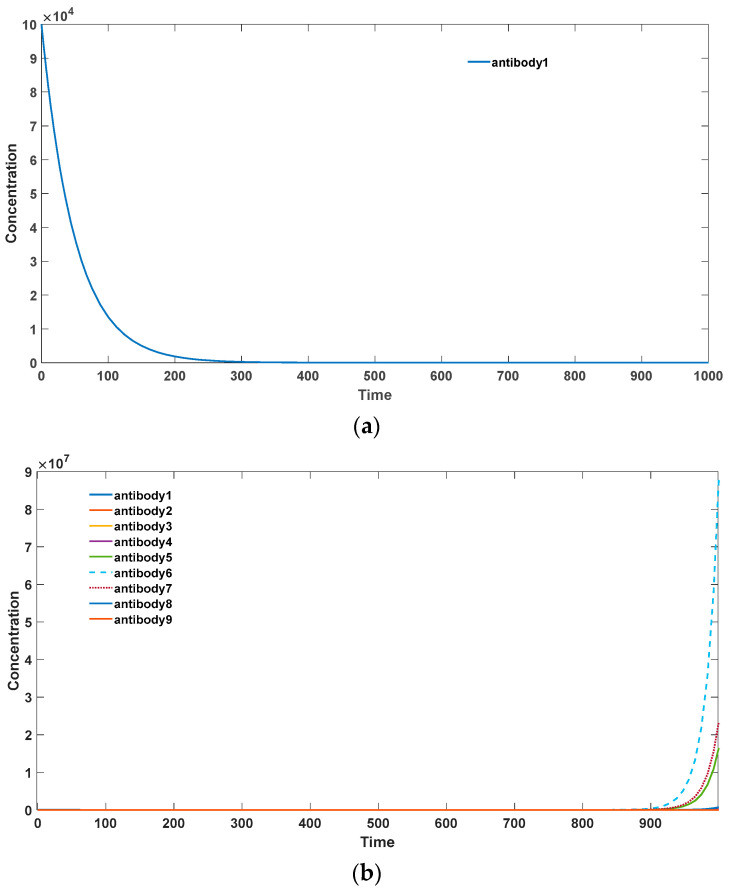
(**a**) Antibody dynamics without SHM in foreign antigen environments. (**b**) Antibody dynamics with SHM rate equal to 0.5 in foreign antigen environments. The parameter sets we used are: *z*(0) = 2 × 10^6^,
$x_{1}\left(0\right)\;$ = 0, $x_{2}\left(0\right)\;$ = 0, $x_{3}\left(0\right)\;$ = 0, $x_{4}\left(0\right)\;$ = 0, $x_{5}\left(0\right)\;$ = 0, $x_{6}\left(0\right)\;$ = 0, $x_{7}\left(0\right)\;$ = 0, $x_{8}\left(0\right)\;$ = 0, $x_{9}\left(0\right)\;$ = 0, $y_{1}\left(0\right)\;$ = 0, $y_{2}\left(0\right)\;$ = 0, $y_{3}\left(0\right)\;$ = 0, $y_{4}\left(0\right)\;$ = 0, $y_{5}\left(0\right)\;$ = 0, $y_{6}\left(0\right)\;$ = 0, $y_{7}\left(0\right)\;$ = 0, $y_{8}\left(0\right)\;$ = 0, $y_{9}\left(0\right)\;$ = 10^5^, 
$k_{2\_1}\;$ = 10^−13^, 
$k_{2\_2}\;$ = 10^−12^, 
$k_{2\_3}\;$ = 10^−11^, 
$k_{2\_4}\;$ = 10^−10^, 
$k_{2\_5}\;$ = 10^−9^, 
$k_{2\_6}\;$ = 10^−8^, 
$k_{2\_7}\;$ = 10^−7^, $k_{2\_8}\;$ = 10^−6^, 
$k_{2\_9}\;$ = 10^−5^, 
$k_{-2}\;$ = 10^−18^, $k_{3}\;$ = 0.5, $k_{4}\;$ = 1.1, $k_{6}\;$ = 
0.02, *p* = 0 in (**a**), *p* = 0.5 in (**b**).

### 3.3. Theory of Immuno-Driven Aging

There are currently many theories about aging, such as the free radical theory [40], the telomere attrition theory [41], and programming aging theory [42]. In recent years, through studies in omics and transcriptomics, new characteristics of aging cells have been discovered, such as high expression of cell inflammatory factors [43,44,45]. Many of these chronic inflammations are mediated by the immune system. Different from the traditional immunosenescence theory [21,22], our proposed theory of immune-driven aging aims to explain that it may be difficult for the immune system to maintain its homeostasis, and over time, the types, composition, and proportions of antibodies will undergo significant changes. The immune system tends to produce more autoantibodies with strong self-binding affinity. As these autoantibodies accumulate, they can lead to various chronic inflammations, which in turn accelerate cellular aging, resulting in the release of a greater amount of self-antigens and more severe inflammatory reactions, ultimately leading to organismal death. Many experiments have shown that antibodies in the human body undergo continuous changes over time. For example, peptide microarray tests for the antibody-peptide binding strength in serum have shown a significant increase in antibody binding to certain specific sequences in the elderly population, while a noticeable decrease in binding strength to other peptides [46]. Somatic hypermutation may be one of the reasons for the evolution of the antibody repertoire, as experiments have indicated that sequence changes brought about by somatic hypermutation are not random but subject to selective pressure [36,37,38]. This also validates our earlier theory of antibody kinetics, namely the role of autoantigens in maintaining memory cells [24]. Antibodies evolve towards a higher binding affinity to autoantigens. To further elucidate this theory, we used mathematical simulations to model the changing trends of antibody composition over time, taking into account the influence of somatic hypermutation. We continue to use our Model 2, and the initial composition and proportion of antibodies released into peripheral tissues is a key consideration. We assume that the initial composition of antibodies follows a normal distribution as described in our Model 2, and we simulate changes in antibody total quantity and composition proportions under external environmental pressures. We test the characteristics of antibody changes at different somatic hypermutation frequencies.

In Figure 6 it can be observed that somatic hypermutation in somatic cells can prevent the excessive accumulation of autoantibodies. When the mechanism of somatic hypermutation is completely inhibited, antibodies with a high binding activity to self-antigens undergo rapid and abundant proliferation, leading to the onset of autoimmune diseases. This point has been well demonstrated by the knockout experiments of AID enzyme in mice, where mice lacking the AID enzyme exhibited strong autoimmune diseases at a young age [39]. Elevating the frequency of somatic hypermutation in an appropriate range can effectively reduce the accumulation of high-affinity autoantibodies while controlling the overall accumulation of antibodies.

Figure 7 illustrates the alterations in the proportions of different antibody types under varying rates of somatic hypermutation in somatic cells. It is evident from Figure 7 that somatic hypermutation serves not only to impede the excessive accumulation of autoantibodies but also exerts an influence on the overall composition of antibodies. Complete abrogation of somatic hypermutation leads to the swift evolution of autoantibodies towards heightened binding affinity. Somatic hypermutation acts as an inhibitory force against this evolutionary progression. Nevertheless, it is worth noting that the trend of antibody evolution towards enhanced self-reactivity appears refractory to reversal, as evidenced by a rightward shift in the distribution of antibody constituents relative to the initial state. Moreover, with the passage of time, the proportion of antibodies exhibiting potent binding affinity to self-antigens progressively augments (as depicted in Figure 8), giving rise to an irreversible alteration in antibody composition. This phenomenon potentially underlies pivotal factors contributing to organismal senescence and the emergence of geriatric chronic inflammation.

The primary content illustrated in Figure 8 pertains to the temporal changes in the proportion of antibodies within the organism. Antibodies exhibiting a strong binding affinity to self-antigens are observed to constitute a higher proportion, resulting in a right-skewed overall distribution as depicted in the figure. In the case of different antibodies, specifically antibodies 1 to 9, their respective affinities to self-antigens progressively increase. Initially, these antibodies originate from newly generated B-cells and have not yet undergone environmental selection. As a consequence, the overall distribution adheres to a normal distribution, characterized by a prevalence of antibodies with intermediate binding affinity, while those with high or low affinities remain less prevalent. With the passage of time, this distribution undergoes a shift, wherein the proportion of high-affinity antibodies gradually escalates. This phenomenon encompasses antibodies 7, 8, and 9. However, due to the presence of somatic hypermutation in somatic cells, it is plausible that antibody 7 exhibits the highest proportion within the depicted figure. Antibodies 8 and 9 possess a considerable likelihood of transforming into antibodies with diminished binding affinities, thereby impeding their capacity to dominate the distribution.

The evolutionary inclination of antibodies towards increased affinity to self-antigens stems from the reciprocal interactions between antigens and antibodies. The regeneration and proliferation of antibodies necessitate stimulation by antigen–antibody complexes. High-affinity antibodies engender a greater generation of antigen–antibody complexes vis-à-vis competition with low-affinity antibodies, thus triggering the production of additional cognate antibodies. Consequently, this drives the evolutionary trajectory of antibodies towards a heightened binding affinity to self-antigens. Notably, the existence of somatic hypermutation in somatic cells serves to temper this process, as hypermutation events occurring in high-affinity antibodies manifest a substantial probability of favoring lower-affinity outcomes. This mechanism ensures that antibodies with exceedingly strong affinities, such as antibodies 8 and 9, do not assert dominance within the distribution.

In order to better depict the evolutionary trend of antibodies, we employed self-antigens that have a strong stimulatory effect on antibody regeneration. Specifically, we increased the value of parameter $k_{4}$ to 0.7. However, it is important to note that this value remains significantly smaller than the stimulating effects induced by exogenous antigens. For instance, in Figure 5b, this value equals 1.1. We used a total antibody concentration (*y*(0)) of 10^6^, following a normal distribution as outlined in Figure 2. From this distribution, we calculated the initial concentrations for each isotype.

## 4. Discussion

Deciphering the mechanisms underlying self-non–self discrimination in the immune system has remained a formidable challenge. It has been established that gene rearrangement and somatic hypermutation within lymphocyte B-cells give rise to a vast arsenal of antibodies, forming an expansive repertoire capable of countering the invasion of diverse pathogenic microorganisms. Nonetheless, the indispensability of self-antigenic substances for the ontogeny and homeostasis of immune cells has progressively come to light. Consequently, a perplexing paradox emerges: how do self-antigens foster the development of B-cells while concurrently excessive affinity engagement provokes their swift demise through clonal deletion? Analogously, the maturation process of T-cells entails seemingly contradictory mechanisms of positive selection and negative selection.

Although researchers have been striving to uncover novel physical mechanisms underlying these discrepancies, scattered specific mechanisms have been discovered in recent years. Unfortunately, none of these mechanisms comprehensively explain the formation of positive selection and negative selection. Therefore, we aim to quantitatively study this process by mathematically modeling the immune response. We have observed that for specific immune responses, there exists a phenomenon where quantitative changes can lead to qualitative changes. Our model has two main characteristics. Firstly, the binding between antibodies and antigens is represented as a dynamic kinetic process, with the kinetic parameters associated with their affinity. Secondly, we explicitly represent the regenerative role of Th cells on B-cells. Concurrently, we recognize that this process is a key factor contributing to the differential induction of immune responses to self-antigens and foreign antigens. Self-antigenic substances can stimulate Th cells to secrete cytokines and other mediators, promoting the proliferation of interacting B-cells and self. Although this stimulation is relatively weak, it facilitates the development of specific B-cells and maintains their certain concentration levels during the maturation process. The same mechanism applies to Th cells. Th cells also undergo somatic recombination, generating various isotypes, with different isotypes encoding distinct TCR sequences that exhibit varying affinities for peptides presented by APCs. When B-cells bind antigenic substances, the antigens are degraded into peptide forms and presented to Th cells via MHC molecules. Th cells with a stronger binding affinity can engage, secrete cytokines, promote B-cell proliferation, and simultaneously trigger their own division and proliferation. Consequently, self-antigenic substances also sustain the early development of Th cells based on the same principles.

Our model suggests that stimulation from self-antigens leads to the rapid clearance of high-affinity autoantibodies, which is beneficial for the biological function of the organism. When an antibody exhibits strong binding affinity to a self-antigen, it rapidly forms a complex with its antigen. However, due to the weak stimulatory effect of peptide sequences generated from the degradation of self-antigens on Th cells, the secretion of cytokines by Th cells is insufficient to counterbalance the clearance rate of antibody-antigen complexes induced by antigen binding. As a result, autoantibodies with a strong binding affinity are quickly cleared, demonstrating a phenomenon known as clonal clearance. This clearance process applies to both B-cells and their corresponding Th cells. In contrast, exogenous antigens elicit significant proliferation of antibodies with strong affinity when invading the body. The main reason for this differential response is the strong T-cell immunogenicity of exogenous antigens, which stimulates the secretion of cytokines by Th cells and promotes the proliferation of corresponding B-cells. From a kinetic perspective, the proliferation effect surpasses the antibody depletion caused by antigen binding, leading to the abundant expansion and selection of high-affinity antibodies.

Our research has shed further light on the profound impact of somatic hypermutation on the immune system, a fascinating and essential aspect of immunology. Distinct from immune cell gene rearrangement, somatic hypermutation is a continuous process that plays a pivotal role in shaping the adaptive immune response. In recent years, an increasing body of evidence has emerged, underscoring the significant influence of somatic hypermutation on the evolution of the human antibody repertoire. The intricate interplay between the immune system and somatic hypermutation has been elucidated through numerous experimental studies. Notably, these investigations have demonstrated that somatic hypermutation not only expands the diversity of antibodies to effectively combat exogenous infections but also contributes to the crucial maintenance of self-tolerance within the immune system. Our mathematical model, aligning with empirical findings, further elucidates these intriguing dynamics. It reveals that somatic hypermutation serves as an intricate process capable of accelerating the clearance of antibodies exhibiting robust binding affinity to self-antigens. This highly regulated mechanism ensures the swift removal of potentially harmful autoantibodies, thereby safeguarding the overall harmony and functionality of the immune system. Conversely, in response to pathogenic incursions, somatic hypermutation facilitates the generation of a plentiful supply of high-affinity antibodies through accelerated antigen-driven selection. This rapid diversification enhances the immune system’s ability to recognize and neutralize invading pathogens skillfully. The multifaceted phenomenon of somatic hypermutation exemplifies the elegance and adaptability of the immune system to generate diverse repertoires of antibodies. Its dual role in both bolstering immune defense against exogenous threats and maintaining self-tolerance highlights the intricacies of immunological responses. Continued exploration of somatic hypermutation holds immense potential for deepening our understanding of immune regulation and fostering innovative approaches for therapeutic interventions in various immune-related disorders.

In addition, building upon this model, we have put forth the hypothesis of immune system-driven aging. We posit that the dynamic characteristics of antibodies contribute to the inability of the immune system’s antibody repertoire to maintain equilibrium. The composition and proportion of antibodies continuously undergo changes, driven by endogenous antigenic stimuli. Broadly speaking, the evolution of antibodies within the organism gradually favors enhanced self-antigen binding affinity. Somatic hypermutation may partially mitigate this process but is insufficient to reverse the trend. Consequently, as age advances, there is a progressive increase in the relative abundance of self-reactive antibodies within the immune repertoire. This surge prompts a heightened self-immune response, ultimately disrupting the homeostasis of other systems. The resulting chronic inflammatory state accelerates cellular apoptosis and even tumorigenesis. Notably, the delicate balance between cell regeneration and apoptosis in various systems, such as epithelial and neural tissues, becomes perturbed. This theory presents a departure from conventional perspectives on immunosenescence, offering insights into the intricate interplay between the immune system and aging processes. Further research in this area holds promise for elucidating the mechanisms underlying age-related immune dysfunction and may pave the way for novel therapeutic strategies targeting immune-mediated aging.

Our research endeavors shed light on the importance of constructing mathematical models, particularly when dealing with complex life phenomena driven by the interplay of multiple factors. Due to disparities in the initial conditions, experimental findings often yield conflicting conclusions. For instance, the transition between positive and negative selection may perplex researchers due to the absence of a straightforward switch mechanism governed by specific regulatory principles.

Nevertheless, it is imperative to recognize a fundamental premise: that all models, by their very nature, are fallible, albeit they possess varying degrees of utility. It is evident that the immune system possesses a remarkably intricate and heterogeneous nature. Regrettably, we have allocated limited attention to the facets of cellular immunity, primarily directing our scientific inquiries towards humoral immunity. Furthermore, we have regrettably disregarded the pivotal influence exerted by regulation T-cells concerning immunosuppression. It is crucial to recognize that our theoretical framework remains subject to various sources of uncertainty, rendering it arduous to provide exhaustive mathematical analyses for multivariate nonlinear systems of ordinary differential equations. As a result, the entire process heavily relies on simulation, and the reliability of numerous parameters necessitates further experimental validation. Exceeding reasonable thresholds in parameter choices may lead to divergent conclusions. Consequently, future research should prioritize additional experimental scrutiny to examine theories such as the non-equilibrium theory of the immune system and its contribution to hastening the aging process.

## Figures and Tables

**Figure 1 biomedicines-11-02048-f001:**
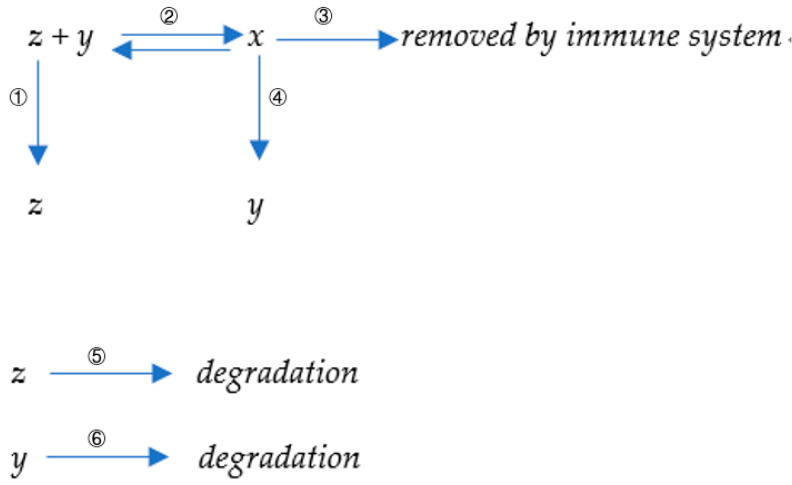
A diagram of antibody-virus interaction.

**Figure 2 biomedicines-11-02048-f002:**
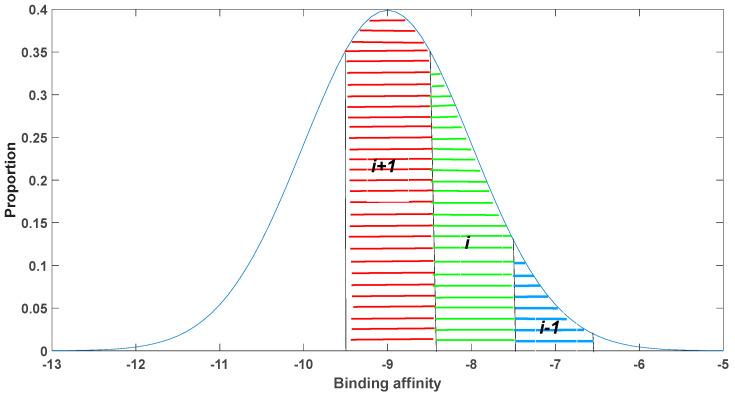
A discretization approach in somatic hypermutation modeling.

**Figure 6 biomedicines-11-02048-f006:**
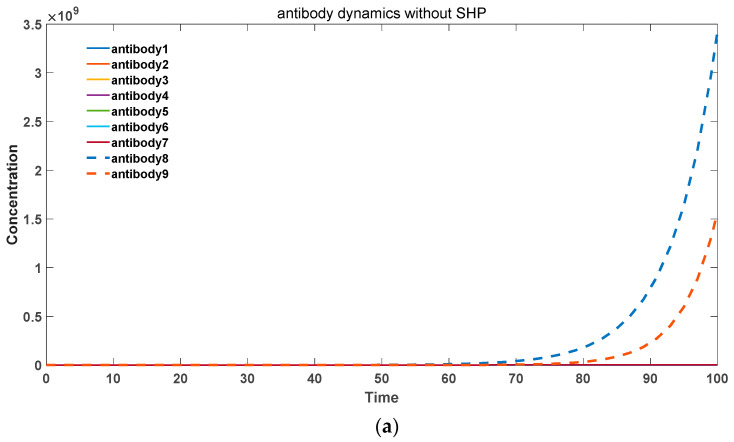
(**a**) Antibody dynamic without somatic hypermutation. (**b**) Antibody dynamic with a somatic hypermutation rate equal to 0.2. (**c**) Antibody dynamic with a somatic hypermutation rate equal to 0.5. The parameter sets we used are: *z*(0) = 2 × 10^6^,
$x_{1}\left(0\right)\;$ = 0, $x_{2}\left(0\right)\;$ = 0, $x_{3}\left(0\right)\;$ = 0, $x_{4}\left(0\right)\;$ = 0, $x_{5}\left(0\right)\;$ = 0, $x_{6}\left(0\right)\;$ = 0, $x_{7}\left(0\right)\;$ = 0, $x_{8}\left(0\right)\;$ = 0, $x_{9}\left(0\right)\;$ = 0, $y_{1}\left(0\right)\;$ = 0.0002 × 10^6^, 
$y_{2}\left(0\right)\;$ = 0.006 × 10^6^, 
$y_{3}\left(0\right)\;$ = 0.0606 × 10^6^, 
$y_{4}\left(0\right)\;$ = 0.2417 × 10^6^, 
$y_{5}\left(0\right)\;$ = 0.3829 × 10^6^, 
$y_{6}\left(0\right)\;$ = 0.2417 × 10^6^, 
$y_{7}\left(0\right)\;$ = 0.0606 × 10^6^, 
$y_{8}\left(0\right)\;$ = 0.006 × 10^6^, 
$y_{9}\left(0\right)\;$ = 0.0002 × 10^6^, $k_{2\_1}\;$ = 10^−13^, 
$k_{2\_2}\;$ = 10^−12^, 
$k_{2\_3}\;$ = 10^−11^, 
$k_{2\_4}\;$ = 10^−10^, 
$k_{2\_5}\;$ = 10^−9^, 
$k_{2\_6}\;$ = 10^−8^, 
$k_{2\_7}\;$ = 10^−7^, 
$k_{2\_8}\;$ = 10^−6^, 
$k_{2\_9}\;$ = 10^−5^, 
$k_{-2}\;$ = 10^−18^, 
$k_{3}\;$ = 0.5, $k_{4}\;$ = 0.7, $k_{6}\;$ = 0.02, *p* 
= 0 in (**a**), *p* = 0.2 in (**b**), *p* = 0.5 in (**c**).

**Figure 7 biomedicines-11-02048-f007:**
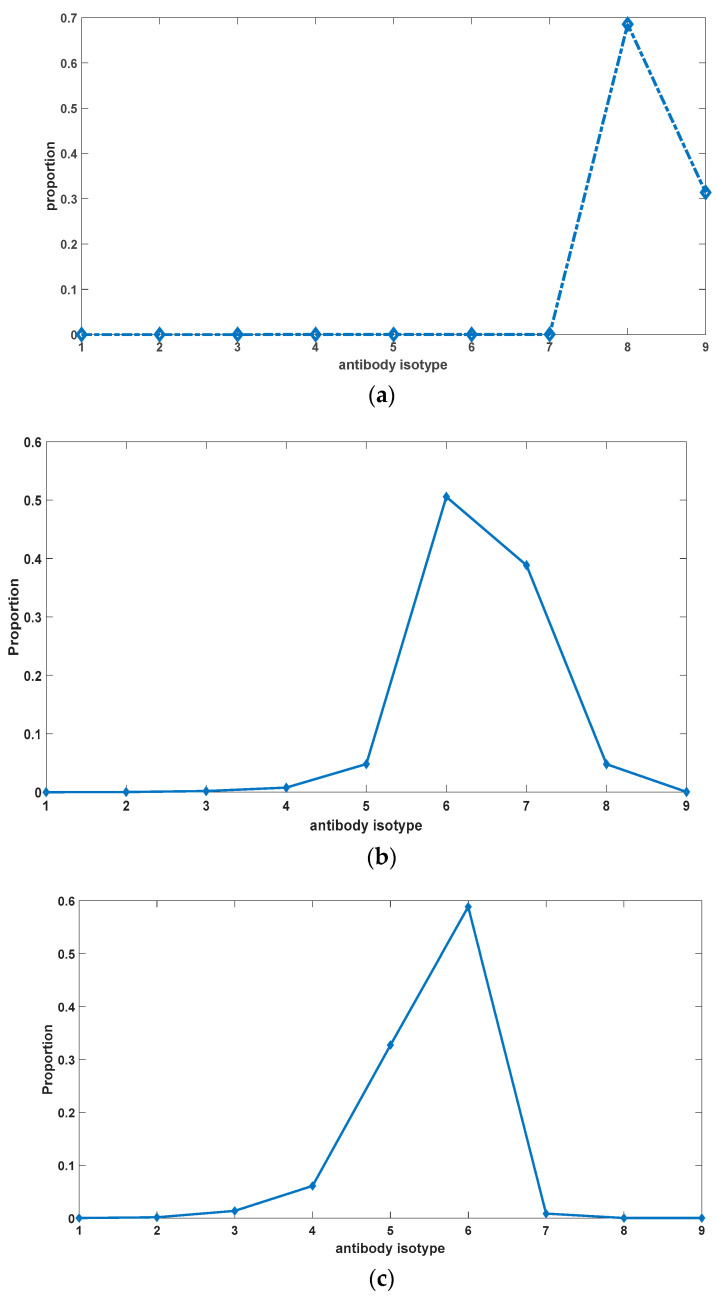
(**a**) Antibody composition without somatic hypermutation. (**b**) Antibody composition with a somatic hypermutation rate equal to 0.2. (**c**) Antibody composition with a somatic hypermutation rate equal to 0.5. The parameter sets we used are: *z*(0) = 2 × 10^6^,
$x_{1}\left(0\right)\;$ = 0, $x_{2}\left(0\right)\;$ = 0, $x_{3}\left(0\right)\;$ = 0, $x_{4}\left(0\right)\;$ = 0, $x_{5}\left(0\right)\;$ = 0, $x_{6}\left(0\right)\;$ = 0, $x_{7}\left(0\right)\;$ = 0, $x_{8}\left(0\right)\;$ = 0, $x_{9}\left(0\right)\;$ = 0, $y_{1}\left(0\right)\;$ = 0.0002 × 10^6^, 
$y_{2}\left(0\right)\;$ = 0.006 × 10^6^, 
$y_{3}\left(0\right)\;$ = 0.0606 × 10^6^, 
$y_{4}\left(0\right)\;$ = 0.2417 × 10^6^, 
$y_{5}\left(0\right)\;$ = 0.3829 × 10^6^, 
$y_{6}\left(0\right)\;$ = 0.2417 × 10^6^, 
$y_{7}\left(0\right)\;$ = 0.0606 × 10^6^, 
$y_{8}\left(0\right)\;$ = 0.006 × 10^6^, 
$y_{9}\left(0\right)\;$ = 0.0002 × 10^6^, $k_{2\_1}\;$ = 10^−13^, 
$k_{2\_2}\;$ = 10^−12^, 
$k_{2\_3}\;$ = 10^−11^, 
$k_{2\_4}\;$ = 10^−10^, 
$k_{2\_5}\;$ = 10^−9^, 
$k_{2\_6}\;$ = 10^−8^, 
$k_{2\_7}\;$ = 10^−7^, 
$k_{2\_8}\;$ = 10^−6^, 
$k_{2\_9}\;$ = 10^−5^, 
$k_{-2}\;$ = 10^−18^, 
$k_{3}\;$ = 0.5, $k_{4}\;$ = 0.7, $k_{6}\;$ = 0.02, *p* 
= 0 in (**a**), *p* = 0.2 in (**b**), *p* = 0.5 in (**c**).

**Figure 8 biomedicines-11-02048-f008:**
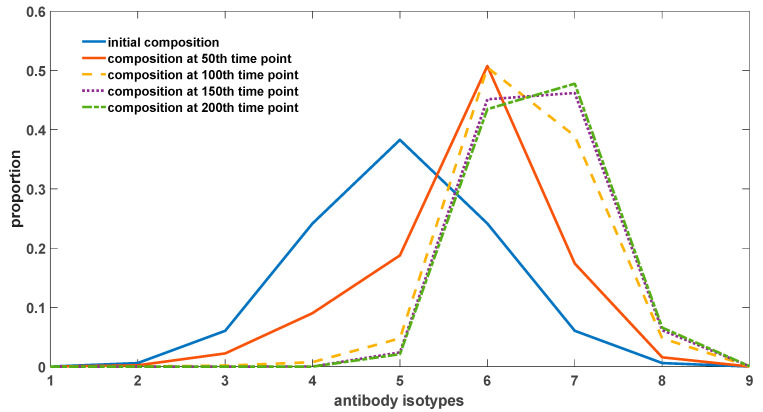
The evolution of antibody composition through time. The parameter sets we used are: *z*(0) = 2 × 10^6^,
$x_{1}\left(0\right)\;$ = 0, $x_{2}\left(0\right)\;$ = 0, $x_{3}\left(0\right)\;$ = 0, $x_{4}\left(0\right)\;$ = 0, $x_{5}\left(0\right)\;$ = 0, $x_{6}\left(0\right)\;$ = 0, $x_{7}\left(0\right)\;$ = 0, $x_{8}\left(0\right)\;$ = 0, $x_{9}\left(0\right)\;$ = 0, $y_{1}\left(0\right)\;$ = 0.0002 × 10^6^, 
$y_{2}\left(0\right)\;$ = 0.006 × 10^6^, 
$y_{3}\left(0\right)\;$ = 0.0606 × 10^6^, 
$y_{4}\left(0\right)\;$ = 0.2417 × 10^6^, 
$y_{5}\left(0\right)\;$ = 0.3829 × 10^6^, 
$y_{6}\left(0\right)\;$ = 0.2417 × 10^6^, 
$y_{7}\left(0\right)\;$ = 0.0606 × 10^6^, 
$y_{8}\left(0\right)\;$ = 0.006 × 10^6^, 
$y_{9}\left(0\right)\;$ = 0.0002 × 10^6^,
$k_{2\_1}\;$ = 10^−13^, 
$k_{2\_2}\;$ = 10^−12^, 
$k_{2\_3}\;$ = 10^−11^, 
$k_{2\_4}\;$ = 10^−10^, 
$k_{2\_5}\;$ = 10^−9^, 
$k_{2\_6}\;$ = 10^−8^, 
$k_{2\_7}\;$ = 10^−7^, 
$k_{2\_8}\;$ = 10^−6^, 
$k_{2\_9}\;$ = 10^−5^, 
$k_{-2}\;$ = 10^−18^, 
$k_{3}\;$ = 0.5, $k_{4}\;$ = 0.7, $k_{6}\;$ = 0.02, *p* 
= 0.2.

## Data Availability

Matlab source codes are available at https://github.com/zhaobinxu23/antibody_dynamics_updated_version, accessed on 1 July 2023.

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
