# Peer review of "Antibody Dynamics Simulation—A Mathematical Exploration of Clonal Deletion and Somatic Hypermutation"

_biomedicines, 2023, doi:10.3390/biomedicines11072048_

Round 1

Reviewer 1 Report

Amazing manuscript, deciphering (with the aid of their mathematical model, enriched in this research), especially mechanisms of clonal deletion and somatic hypermutation (of B cells), with consecutive complex repercussions in immune responses and interactions (including even autoimmunity and self-tolerance), and not only (e.g. immunological aging process in humans). The paper is well written, in an elegant manner. Figures are complete and illustrative. The authors paid attention to details. Discussion paragraph is also nicely conceived, the authors pointing out also the limitations of their approach. I congratulate the authors and support the publications of this work.

Author Response

Thanks a lot for your interest and comments on this research!

Reviewer 2 Report

Dear authors

Title: Antibody Dynamics Simulation— A Mathematical Exploration of Clonal Deletion and Somatic Hypermutation

In this paper, a mathematical modeling techniques to construct a comprehensive framework for elucidating the intricate response mechanisms of the immune system, facilitating a deeper understanding of complex immunological phenomena. Building upon our previous anti- body kinetics theory, we have further enriched this model, incorporating several significant advancements in the field. This enhanced model encompasses multiple key aspects for analyzing immune responses in a more sophisticated and comprehensive manner

Few comments are required to improve the quality of the paper as:

1) Abstract is too long, it should be comprehensive.

2) Provide the organization of the paper

3) From where the numerical values have been taken

4) Provide the reference of the equations

5) In Fig. 7, the graph rises on the antibody isotype 7, why?

6) Provide the suitable methodology 

Author Response

Few comments are required to improve the quality of the paper as:

  • Abstract is too long, it should be comprehensive.

Response: Thanks for your comments. The abstract is revised and condensed.

  • Provide the organization of the paper

Response: A detailed organization of the paper is provided at the end of the introduction part.

  • From where the numerical values have been taken

Response: Detailed explanation and some references to the numerical values are provided.

  • Provide the reference of the equations

Response: The reference of the equations (1-3) is provided. Equations (4-9) are developed in this paper.

  • In Fig. 7, the graph rises on the antibody isotype 7, why?

Response: The primary content illustrated in Figure 7 pertains to the temporal changes in the proportion of antibodies within the organism. Antibodies exhibiting a strong binding affinity to self-antigens are observed to constitute a higher proportion, resulting in a right-skewed overall distribution as depicted in the figure. In the case of different antibodies, specifically antibodies 1 to 9, their respective affinities to self-antigens progressively increase. Initially, these antibodies originate from newly generated B cells and have not yet undergone environmental selection. As a consequence, the overall distribution adheres to a normal distribution, characterized by a prevalence of antibodies with intermediate binding affinity, while those with high or low affinities remain less prevalent. With the passage of time, this distribution undergoes a shift, wherein the proportion of high-affinity antibodies gradually escalates. This phenomenon encompasses antibodies 7, 8, and 9. However, due to the presence of somatic hypermutation in somatic cells, it is plausible that antibody 7 exhibits the highest proportion within the depicted figure. Antibodies 8 and 9 possess a considerable likelihood of transforming into antibodies with diminished binding affinities, thereby impeding their capacity to dominate the distribution.

The evolutionary inclination of antibodies towards increased affinity to self-antigens stems from the reciprocal interactions between antigens and antibodies. The regeneration and proliferation of antibodies necessitate stimulation by antigen-antibody complexes. High-affinity antibodies engender a greater generation of antigen-antibody complexes vis-à-vis competition with low-affinity antibodies, thus triggering the production of additional cognate antibodies. Consequently, this drives the evolutionary trajectory of antibodies towards heightened binding affinity to self-antigens. Notably, the existence of somatic hypermutation in somatic cells serves to temper this process, as hypermutation events occurring in high-affinity antibodies manifest a substantial probability of favoring lower-affinity outcomes. This mechanism ensures that antibodies with exceedingly strong affinities, such as antibodies 8 and 9, do not assert dominance within the distribution. We added the above explanation to the manuscript.

6)  Provide the suitable methodology 

Response: The methodology part has been revised. Section 2.2 and 2.3 are merged, and detailed explanations are also provided, including the corresponding applications of those 2 models.

Reviewer 3 Report

The authors used mathematical models to simulate clonal deletion and somatic hypermutation and the dynamics of antibody formation.

As for the logic, no problems are perceived, especially since the simulation is also a simulation. In addition, it is thought that the understanding of what actually occurs in living organisms will be advanced from the many simulations.

Now, do the authors have any specific examples of the need for a new model that they felt needed to be understood in a new way, as actual data from actual clinical practice?

Or, in order to show the correctness of this mathematical model, it would be even better if you could describe what kind of molecular changes in what kind of cells should be observed from actual human specimens.

Author Response

Thanks a lot for your comments. We do appreciate your interest in this work. The choice of parameters has a profound impact on the simulation outcomes. Typically, parameter values are determined through literature mining or experimental fitting. Regrettably, specific experimental data pertaining to the clonal deletion of B cells and somatic hypermutations in the context of our model remains elusive, thus diminishing the model's veracity. One significant reason for the lack of such data is the challenge in quantifying self-antigenic substances accurately. Unlike exogenous pathogens, self-antigens encompass a wide variety of molecules and exhibit diverse binding affinities with antibodies. Nonetheless, we eagerly anticipate clinical data, particularly regarding autoimmune diseases targeting specific self-molecules. By integrating actual antibody and self-antigen levels, we can effectively fit the data to determine more realistic immunological parameters. We would be honored to collaborate with your group if you possess immunological data or experimental ideas. Utilizing concrete examples rather than simple theoretical models is our future research direction.

Round 2

Reviewer 2 Report

Accept in present form